# The Potential of BIM in Disseminating Knowledge about Decentralized Wastewater Treatment Systems: Learning through the Design Process

**Matheus Alves Dariva** *ⓘ **and André Araujo** ⓘ

Graduate Program in Architecture and Urbanism, School of Architecture and Urbanism, Campus Santa Mônica, The Federal University of Uberlândia, Uberlândia 38408-100, MG, Brazil; andre.araujo@ufu.br
* Correspondence: matheusalvesdariva@gmail.com

**Abstract:** Despite the low rate of sewer service coverage in developing countries, especially in small towns and rural areas, decentralized wastewater treatment systems (DEWATS) are alternatives to ensure public health. Nonetheless, understanding the physical or functional aspects of these system constructions remains difficult when there is no bibliography, a teaching professional, or even a prototype that allows technical conclusions about specific processes. Although resource combination in a BIM environment works as a facilitator to obtain compact and sustainable results, the purpose of this collaboration is not only to simplify the design process. This paper aims to propose a way to use BIM concepts as a learning tool throughout the modeling process. To do so, we developed a BIM template with specific DEWATS characteristics in order to provide not only information needed for construction, but also information that can favor learning. We measured how much the level of development (LOD) can influence learning about these systems during the design process, directing users to make the best implementation choices. The adoption of qualitative analysis of a questionnaire answered by three groups of professionals (general CADD software users, BIM software with generic template users, and BIM software with specific DEWATS template users) allowed us to identify differences among them, such as the number of DEWATS known, considered LOD, learning, and handicaps in the design of these systems. The results pointed out that the BIM tools can influence learning about new and unknown specific systems, which is directly related to the LOD of a model.

**Keywords:** BIM; learning; level of development; decentralized wastewater treatment systems





## 1. Introduction

### 1.1. The Scenario of Basic Sanitation in Developing Countries

Sanitation best practices are fundamental to prevent diseases, especially those transmitted via wastewater, as well as to promote public health, to protect the environment, and to increase the population's quality of life [1]. It is possible to define the sanitary sewer system as the set of activities, infrastructure, and operational facilities for collection, transportation, treatment, and final wastewater disposal, from the building of connections to their final release into the environment [2]. Despite being an essential basic sanitation service for the promotion of public health and environmental protection, around 2.4 billion people, that is, 32% of the global population, still lack adequate sanitation facilities. This situation is even more critical when it comes to developing countries. Since 1990, the number of countries with less than 50% of their population having access to sewer services has declined slightly, from 54 to 47. Overall, sanitation coverage in developing countries comprises only 49% of the population, representing half that of developed countries (98%) [3].

This is also a reality in Brazil, where sewer services reach 53.2% of its population (105.5 million inhabitants), and only 46.3% of the wastewater generated in 2018 was treated. This scenario is even more worrying when analyzing the situation in the north and

northeast regions of Brazil, which had only 21.7% and 36.6% of their wastewater treated, respectively. Throughout Brazilian territory, there was a slight growth in the volume of treated wastewater, which went from 4.18 billion $m^3$ in 2017 to 4.30 billion $m^3$ in 2018, corresponding to only 2.9% of the total amount of wastewater treatment in relation to water consumed in 2018, according to the wastewater treatment index [4]. These data are displayed in detail in Table 1, which was adapted from the 24th Diagnosis of Water and Sewage Services of the Brazilian National Sanitation Information System (SNIS).

**Table 1.** Levels of water and sewage services presented by SNIS service providers in 2018.

| Macro-Region | Network Service Index (%) | | Wastewater Treatment Index (%) | |
|---|---|---|---|---|
| | Total Population with Access to Water Service | Total Population with Access to Sewer Service | Wastewater Treatment in Relation to Water Consumed | Collected Wastewater Treatment |
| North | 57.1 | 10.5 | 21.7 | 83.4 |
| Northeast | 74.2 | 28.0 | 36.2 | 83.6 |
| Southeast | 91.0 | 79.2 | 50.1 | 67.5 |
| South | 90.2 | 45.2 | 45.4 | 95.0 |
| Midwest | 89.0 | 52.9 | 53.9 | 93.8 |
| Brazil | 83.6 | 53.2 | 46.3 | 74.5 |

Source: Adapted from SNIS (2019).

*1.2. Recent Advances in DTW Systems*

Despite the low coverage rate of sewer services in developing countries, especially in small towns and rural areas, decentralized wastewater treatment systems (DEWATS), if well designed, built, and operated, are alternatives to ensure public health and to maintain the environmental integrity of these locations [5] DEWATS are those which collect, treat, and reuse or carry out the final wastewater disposal in a place close to its generation, unlike what happens in traditional centralized systems, where the wastewater from whole cities is collected and treated in large wastewater treatment plants [6].

Regarding the search for universal basic sanitation services, the decentralization strategy has proven to be increasingly complementary to the centralization of wastewater treatment, since the substantial capital investment made in the implementation of centralized systems can be reduced, thus increasing the accessibility of wastewater management systems. However, the lack of research activities in developing countries has led to the selection of inappropriate technologies in terms of local climatic and physical conditions, financial and human resources, and social and cultural acceptance [5], thereby harming the environment and failing to take advantage of some other benefits, such as reuse in agriculture [7] and construction [8]. The first challenge in implementing this type of system is choosing the most appropriate wastewater treatment technologies for each situation—a complex task, which involves evaluating many variables simultaneously. There is now a wide variety of technologies available for DEWATS, but there is no consensus on which would be the most technically adequate [1–6]. The decision must take into account local specificities, given that there are significant differences among different regions of the world, with regard to their climatic, ecosystem, environmental, socioeconomic, and cultural characteristics [3–6].

Although there are a wide range of efficient wastewater treatment systems available, such as constructed wetlands, evapotranspiration tanks, sand filters, anaerobic filters, banana circles, and others, many professionals from architecture, engineering, construction, and operation (hereafter AECO) are not aware of these various possibilities or of the functionality of some equipment [6]. In view of the large system variability, studies have pointed to the existence of a relationship between design and operational parameters and treatment performance [9]. Therefore, there must be a more significant concern in studying

and working efficiently with the information related to this type of system in the design process. The solutions adopted in this stage directly reflect on the construction process and on the final product quality [10,11]. In this scenario, BIM technology can fit not only as a project method, but also as a learning tool that can aid project teams in becoming acquainted with specific systems and construction tasks before the commencement of the task on site [12]. That is, there is a possibility for professionals to learn about DEWATS by designing these systems in a BIM software.

### 1.3. BIM as a Learning Tool

In teaching BIM, the collaborative aspect is commonly pointed out as the central aspect, as it aims to integrate expertise during building construction [13,14]. This type of approach reproduces project teams and, in addition to gathering knowledge, promotes joint learning through an exchange of experiences [15]. The adoption of this strategy can contribute to the training of more qualified office employees, as well as seemingly contributing to the improvement of project information management skills [16]. Nonetheless, understanding physical aspects remains difficult when there is no bibliography, a teaching professional, or even a prototype that allows technical conclusions about a process [17]. The view of learners when establishing comparisons between learning through traditional CAD design and the information control on BIM, in general, is restricted to the design itself within these two systems [18]. The ability to learn a process using either tool is much more challenging to measure, especially when design demands the construction of specific systems.

Although resource combination in a BIM environment works as a facilitator to obtain more compact and sustainable results, the purpose of this collaboration is not only to simplify the design process. Some inferences about the integration of the disciplines— architecture, structure, electrical, and hydraulic—continue leading to sequential linearity among them [19]. The compact synchronization of these specificities remains one of the main obstacles [20]. A research carried out with small samples indicated that users believe that using a BIM application in its architectural version is more effective in learning structural systems than using BIM in its mechanical, electrical, and plumbing (MEP) version to learn MEP systems [21]. One possible interpretation is that users learn more about a system when they have additional prior knowledge about its functionality [22].

Countries' income seems to influence this perception, since many approaches seek to import technological standards from developed countries to developing countries [19]. More work is needed to create new BIM solutions that best suit the context of local construction industries in developing countries [23]. DEWATS, for instance, can be an efficient alternative for the north and northeast regions of Brazil, as these regions have high temperatures, long hours of radiation, availability of land, and a low percentage of sewer coverage [6]. Systems like these require specialized knowledge about the construction process to document it at a development level that is compatible with its constructability. In an ideal context, the professional who develops these solutions does not simultaneously engage in other projects [24]. Moreover, the time and cost of developing specialized solutions with high levels of development are normally increased, but they reflect savings in the construction and management phases [25].

From the brief context outlined above, this paper aims to propose a way to use BIM concepts as a communication and learning tool throughout the modeling process. In this sense, we developed a BIM model with specific DEWATS characteristics in order to provide information that could favor learning. We measured how much the level of development (LOD) can influence learning about these systems during the design process, directing users to make the best implementation choices. The adoption of qualitative strategies and a holistic analysis of a questionnaire answered by three groups of professionals allowed us to identify differences among them. The results pointed out that the BIM tools can influence learning about new and unknown specific systems and help professionals to make best choices for DEWATS implementation.

## 2. Materials and Methods

In order to test the hypothesis, we developed an artifact based on the design science research method. Firstly, we surveyed the relevance of essential sanitation services, which was presented in the previous sections. We concluded that DEWATS can serve as complementary systems to promote public health and local sustainability. However, the scientific community still faces challenges to promote the dissemination of this knowledge, since many professionals are unaware of the vast possibility of existing systems. Therefore, we developed an artifact in order to verify professionals' knowledge about DEWATS. Through our bibliographic survey, it was possible to conclude that BIM technology has the potential to disseminate specific knowledge through different methodologies. Thus, the idea was to develop a specific DEWATS template for BIM software that presents several options of very-high-LOD systems that could guarantee not only information needed for construction, but also information that could favor learning.

The template presents 14 possibilities of DEWATS (Figure 1) that, if well designed and built, guarantee wastewater treatment efficiency [6]. They are a circular septic tank, rectangular prismatic septic tank and anaerobic filter set, evapotranspiration tank, EMBRAPA biodigester septic tank, anaerobic baffled reactor, biodigester, constructed wetlands, banana circle, circular anaerobic filter, chlorination tank, sand filter, vermifilter, circular soak pit, and rectangular soak pit. The systems were designed following rigorous standards and courseware to ensure correct sizing, to meet needs, and to favor learning. To do so, several algorithms were created and applied to several model parameters. All the components were configured with the possibility of evaluating and studying each one separately, as well as with the possibility of self-sizing them and automatically creating quantitative tables, calculation memorials, and didactic information. It is also possible to analyze information by positioning the mouse cursor over a parameter and to identify each one of its elements. The template also presents standard items available for study, as well as references that can be analyzed in the components' parameters. Figure 2 displays how the component parameters work.

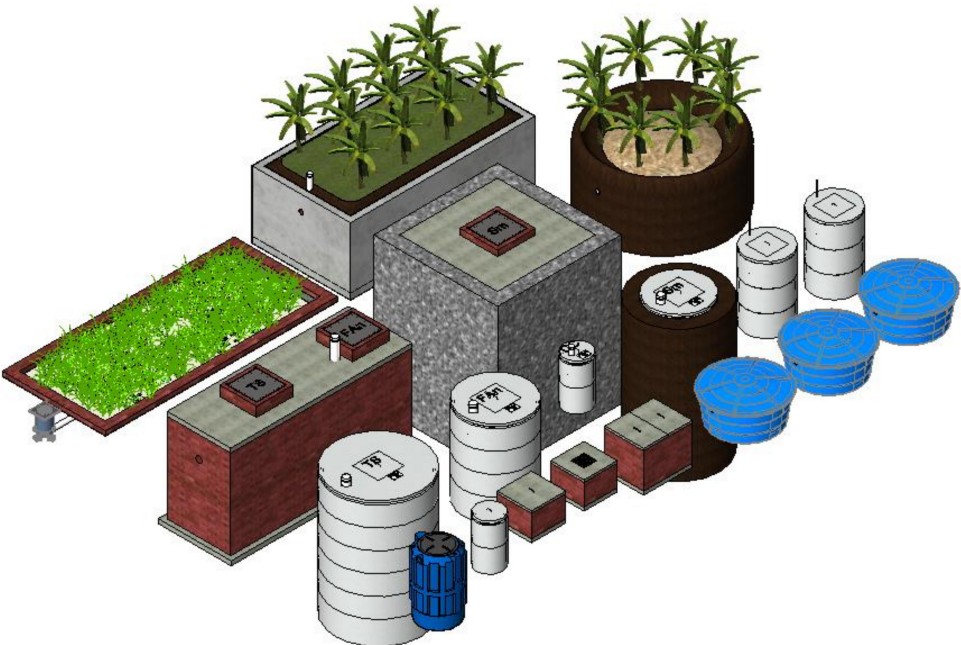

**Figure 1.** DEWATS in the specific template. Source: authors (2021).

Each family was named and configured according to courseware and standards [1,6].

In the parameter group "Construction", the system sizing parameters are available. They are referenced on standards and courseware. Based on input such as "number of people", "retention time" "contribution of evictions", and others, the equipment is automatically sized according to commercial standards.

By positioning the mouse cursor on some parameter, it is possible to analyze additional information, such as parameter meaning, measurement units, reference standards, reference values, and others

Constructive parameters such as number of concrete rings or "pipes diameters" can also be modified, allowing the study of several possibilities.

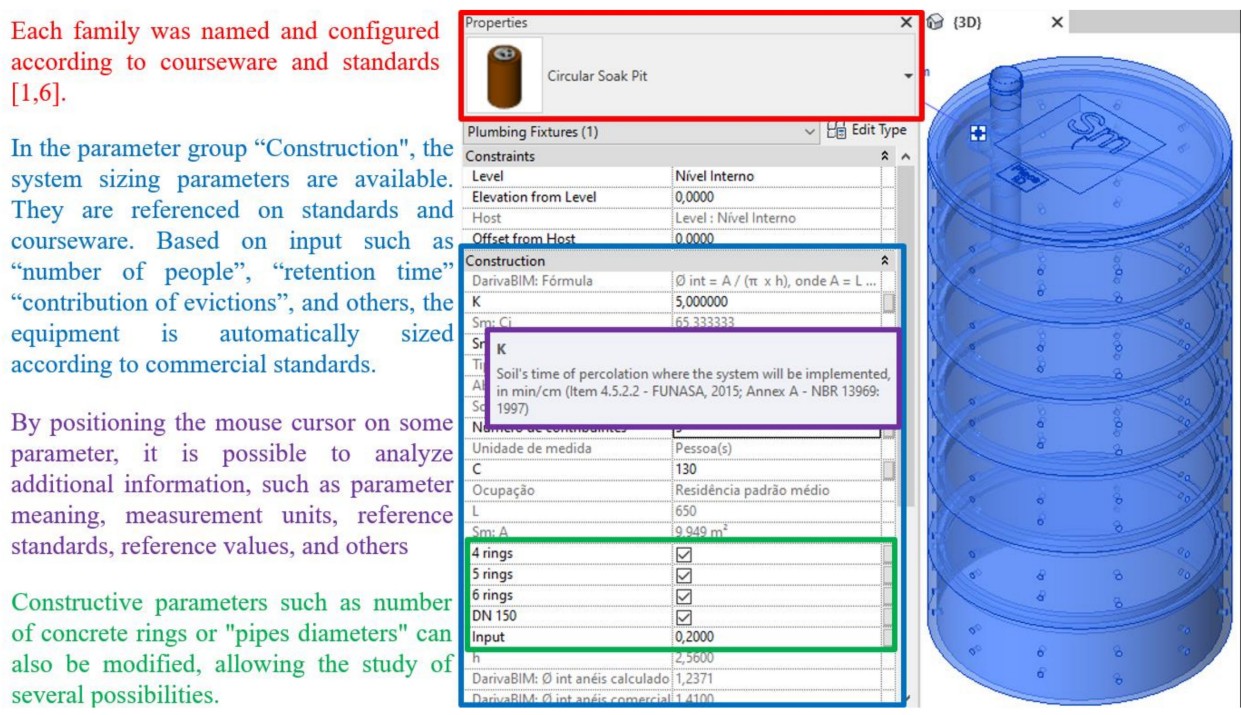

**Figure 2.** Some information about how the component parameters work. Source: authors (2021).

In the template, when one of these DEWATS components is used in the project, several tables are added with information about the systems. Sizing information, system type, application, treatment stage, and organic matter removal are some examples, as we can see in Figure 3. Thus, the purpose of this high level of development is to provide the possibility of learning about unknown systems and direct the users to make the best choices in system implementations for the diverse possibilities of project characteristics.

| DEWATS INFORMATIONS | | | | | | | |
|---|---|---|---|---|---|---|---|
| Component | System type | Application | Treatment stage | Organic matter removal | Maintenance frequency | Sizing standards | Operation and maintenance |
| Circular Septic Tank | Decentralized Wastewater Treatment Systems | Sanitary Basin Sewage (black waters) and Domestic Sewage (gray waters) | Initial | Moderate | Low | ABNT. BRAZILIAN ASSOCIATION OF TECHNICAL STANDARDS. NBR 7229: Design, construction and operation of septic tank systems. Rio de Janeiro: Abnt, 1993. 15 p. | Sludge and scum will have to be removed! Leave approximately 10% of its volume inside the tank (Item 6.2.1.3 - NBR 7229: 1993). There are two options for cleaning: - Local sludge management (use Drying Bed to treat the sludge); - Sludge removal by clean pit truck (outsourced labor). The cleaning interval is described in the septic tank memorial. |
| Circular Anaerobic Filter | Decentralized Wastewater Treatment Systems | Pre-treated Sewage in the Initial Stage (it is recommended that it is preceded by a Septic Tank, Biodigester or Anaerobic Reactor, Compartmented) | Filtration | Moderate | Low | ABNT. BRAZILIAN ASSOCIATION OF TECHNICAL STANDARDS. NBR 13969: Septic tanks - Complementary treatment units and final disposal of liquid effluents - Design, construction and operation. Rio de Janeiro, 1997. 60 p. | The sludge will have to be removed! There are two options: - Local sludge management (use Drying Bed to treat the sludge); - Sludge removal by clean pit truck (outsourced labor). Cleaning should be carried out when the filter bed is obstructed. |

**Figure 3.** Automatically generated table of information about the DEWATS used in the projects. Source: authors (2021).

*Artifact Evaluation and Contribution*

The scientific reduction worked with a holistic approach to our samples, in order to interpret the results in natural settings, without extracting them from their contexts. On the basis of inductive logic, which assumes the influence of modeling tools on learning, hypotheses were formulated, considering the limitations of our scientific method. However, despite being careful in terms of result generalization, the extended contact with the samples allowed us to identify common and specific characteristics. The inferences presented in

this paper are outcomes from questionnaire analysis and BIM model assessments. The qualitative experimental strategy was used for the analysis of the questionnaires and artifact assessments. The questionnaire was designed and distributed to 75 professionals in order to assess their point of view on DEWATS according to their design process background. The first question was as follows: "What is the design tool that you use in the conception of piping and DEWATS projects?" This question was aimed at dividing these professionals into three groups of 25 people each:

- Treatment 1 = General computer-aided design and drafting (CADD) software users (33.33%);
- Treatment 2 = BIM software with generic template users (33.33%);
- Treatment 3 = BIM software with specific DEWATS template users (33.33%).

  The next six questions had the following configuration:

- Second question: How many types of DEWATS do you know?
- Third question: How do you document DEWATS in projects?
- Fourth question: In your documentation method, how do you classify the role of the LOD in helping you comprehend budgeting, construction, maintenance, and operating of the systems?
- Fifth question: Do you learn about DEWATS throughout the design process?
- Sixth question: For you, what are the main handicaps when representing DEWATS?
- Seventh question: From your point of view, does the number of available BIM templates for sale generate resistance to change from the traditional drafting CAD project method to BIM?

Treatment 1 was the control group, which was the condition used to compare values obtained in this research. The objective was to measure if information-based modeling can be related to learning in the design process and if a model's LOD increase can favor learning. Through the analysis of the answers to the questionnaire, we were able to draw some conclusions about this linkage. The results obtained are presented and discussed in Sections 3 and 4; however, they should be interpreted with some caution, avoiding generalizations, as they are one of possible interpretations. They also can be viewed through the following link: https://docs.google.com/forms/d/1pM30ZzYOTozXQMbfhzfh-oSp7 cqohcWGBx4_FAKg4BM/viewanalytics (accessed on 5 May 2021).

## 3. Results

For a more in-depth analysis of the results, it was observed how each treatment group (general CADD software users, BIM software with generic template users, and BIM software with specific DEWATS template users) answered the other questionnaire questions (from the second to seventh questions). The graphs in this section display the relationships underlying the answers of the professionals from the three groups. These relationships are presented in Figures 4–9.

### 3.1. Known Systems

The intention of the second question was to carry out an average survey of the number of systems known by each group of professionals. With this information, it was possible to establish some relationships with the answers to other questions. The average number of systems known by each group allowed us to make conjectures about learning unknown systems in different design methods. These relationships were established in the discussion.

Second question's graph indications (Figure 4): If a group of professionals knew 1–3 systems, they were considered to know a small number of systems. If they knew 4–6 systems, they were considered to know a reasonable number of systems. If they knew 7–9 systems, they were considered to know a good number of systems. Lastly, if they knew 10 or more systems, they were considered know many systems. Despite the subtle difference, it is possible to notice that general CADD software users tended to know fewer systems when compared to BIM software with generic template users, among which 80%

also knew a small or reasonable number of systems. Comparing these two groups with the BIM software with specific DEWATS template users, we can notice a clear difference. Despite presenting a more comprehensive distribution among the professionals of this third group, most of them (60%) knew a high number of DEWATS.

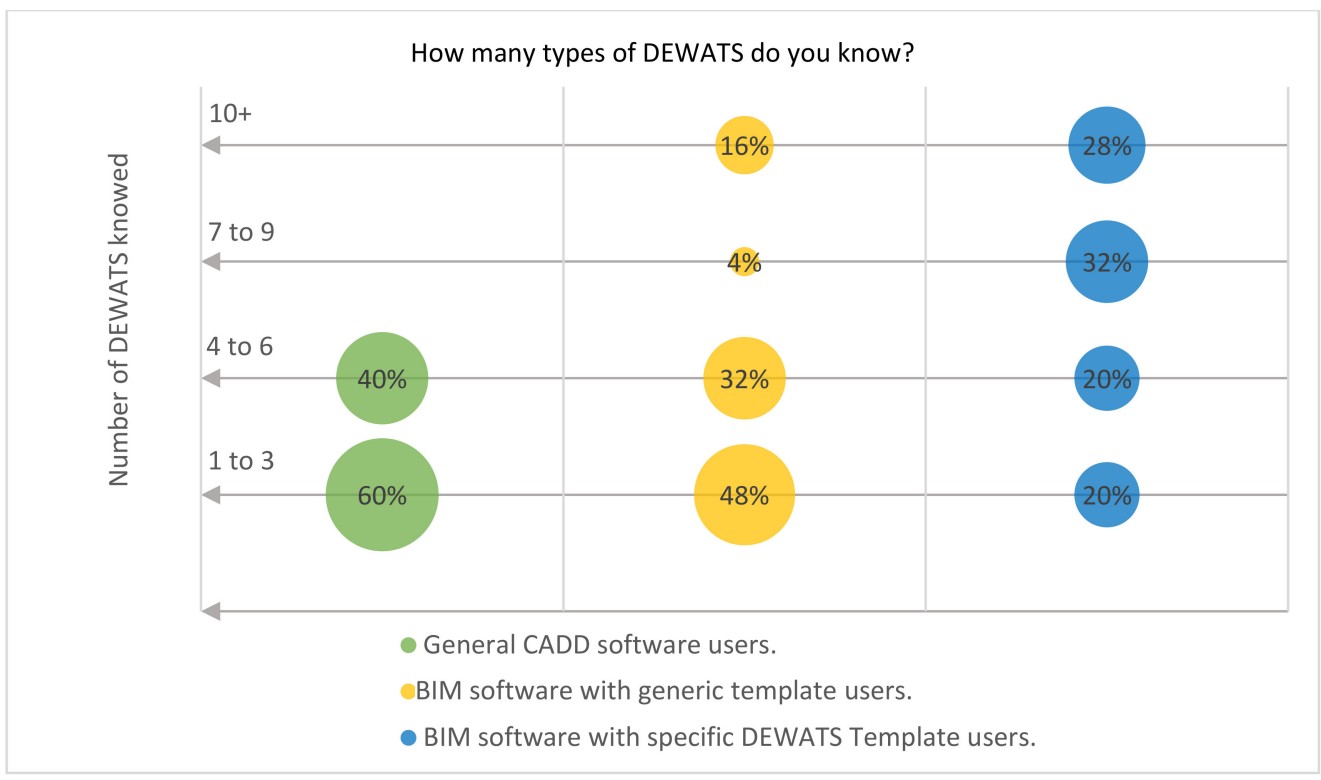

**Figure 4.** Answers to the second question. Source: authors (2020).

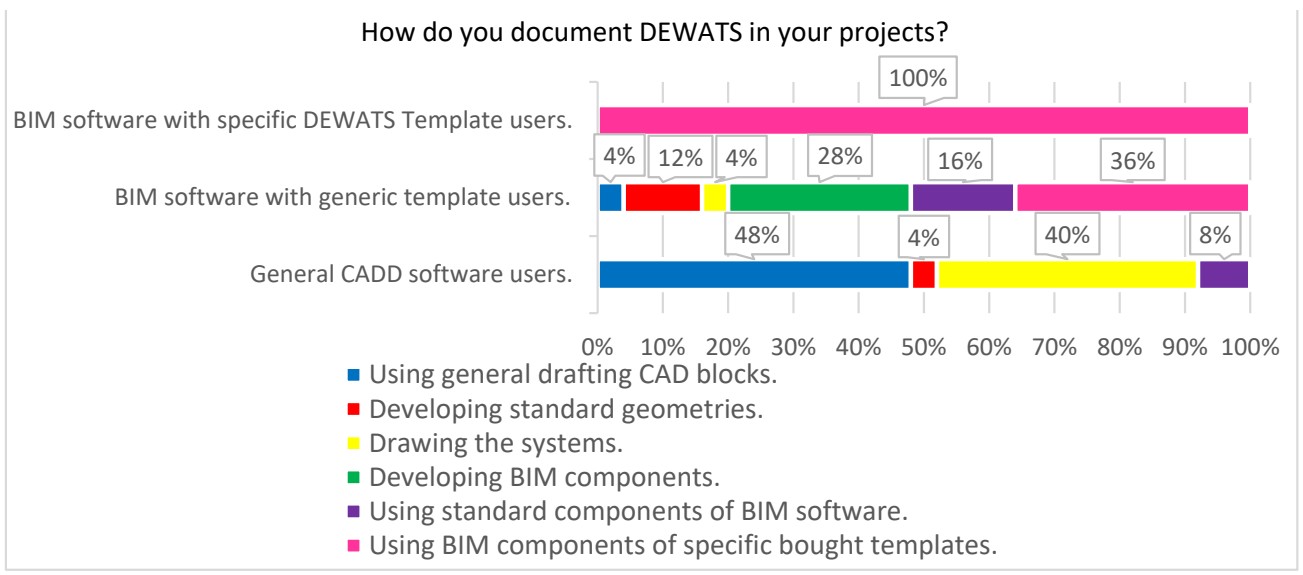

**Figure 5.** Answers to the third question. Source: authors (2020).

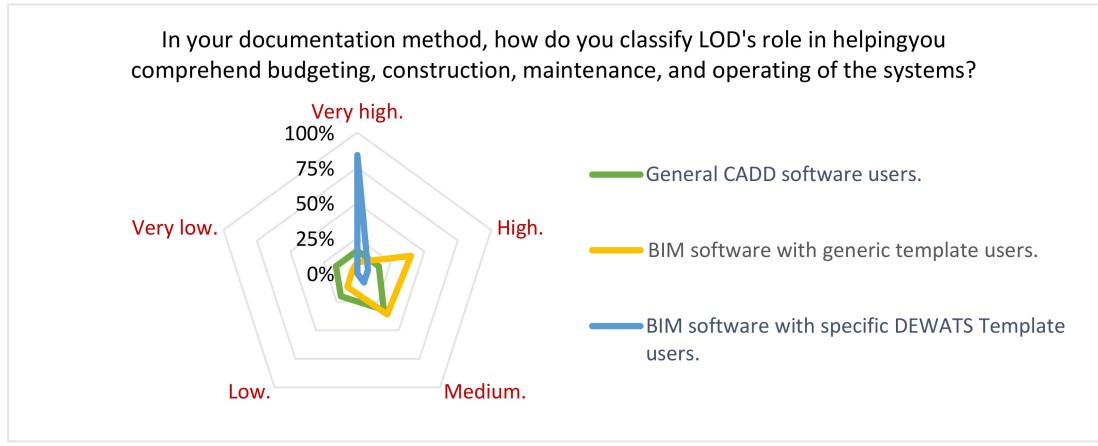

**Figure 6.** Answers to the fourth question. Source: authors (2020).

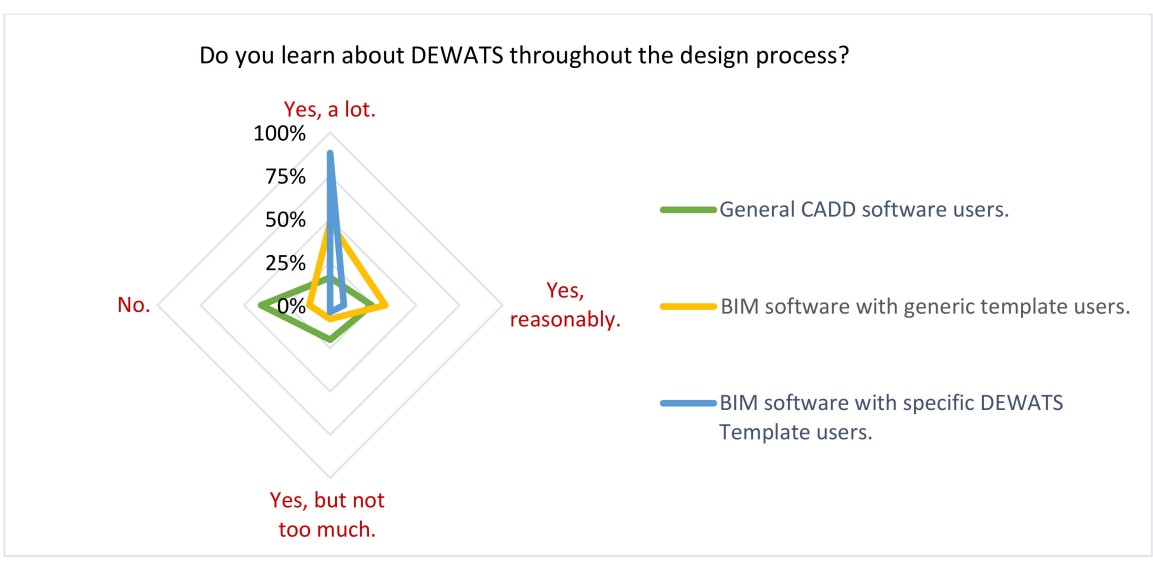

**Figure 7.** Answers to the fifth question. Source: authors (2020).

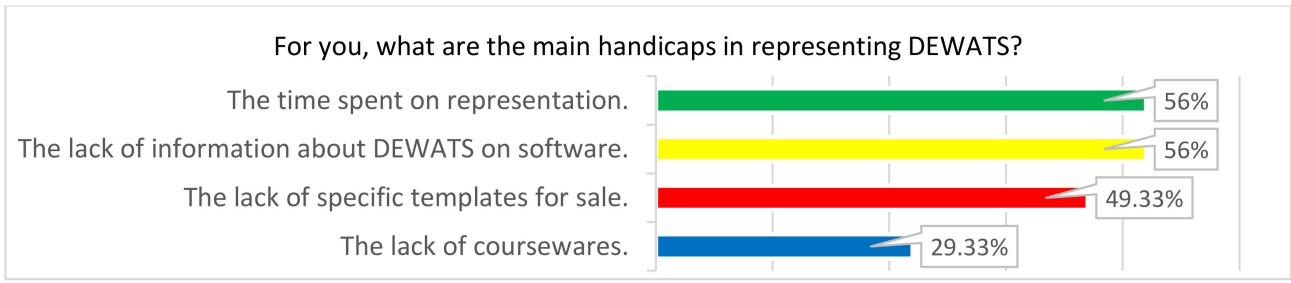

**Figure 8.** Answers to the sixth question. Source: authors (2020).

### 3.2. DEWATS Documentation Method

It was considered relevant to identify the DEWATS documentation method used by each professional, enabling a verification of whether the professional used the design tool itself to document these systems. The answers to question 3 demonstrate this information.

Third question's graph indications (Figure 5): Almost all (88%) general CADD software users used general drafting CAD blocks or drew DEWATS representations in projects. BIM software with generic template users, on the other hand, explored several methods to represent these systems in projects, but the prevailing methods were the use of BIM

components with specifically bought templates, BIM components developed by them, or standard components of BIM software. However, 20% of these professionals still developed standard geometries, drew the systems, or used general drafting CAD blocks. Lastly, BIM software with specific DEWATS template users unanimously used BIM components of specifically bought templates.

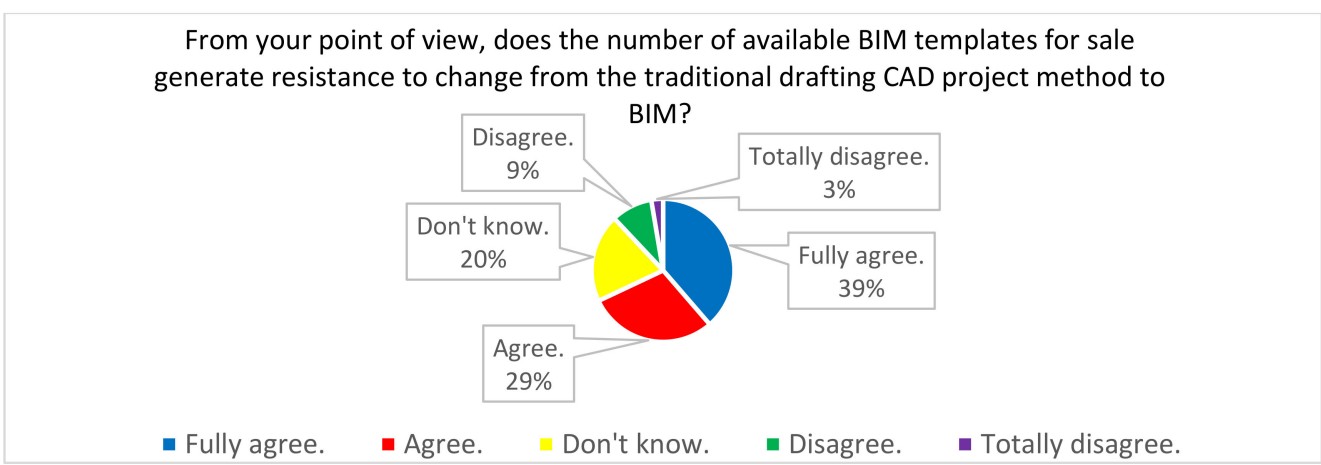

**Figure 9.** Answers to the seventh question. Source: authors (2020).

### 3.3. Considered LOD

The fourth question was aimed at identifying how each professional classified the LOD of their DEWATS model in their documentations method. The intention of this question was to establish a relationship with the fifth question, i.e., between LOD and level of learning.

Fourth question's graph indications (Figure 6): The LOD classification for DEWATS representations in projects by the majority of CADD software users was well distributed, with a tendency toward "medium" or "low". For BIM software with generic template users, the LOD classification of the systems tended to be "high" or "medium". Lastly, BIM software with specific DEWATS template users revealed a typical classification of "very high".

### 3.4. Considered Learning

With the fifth question, it was possible to identify how much the professionals considered that they learned about DEWATS in their documentations method in the design process. The intention was to verify if a BIM template designed in high LOD with specific elements of DEWATS could contribute in some way to favor learning. This relationship was analyzed in the discussion.

Fifth question's graph indications (Figure 7): Despite presenting a more comprehensive distribution, 60% of general CADD software users did not learn with this design tool or did not learn much about DEWATS throughout the design process. The majority (80%) of BIM software with generic template users learned a reasonable amount. Lastly, the majority (88%) of BIM software with specific DEWATS template users learned substantially about DEWATS throughout the design process.

### 3.5. Handicaps in Representing DEWATS

The sixth question was intended to identify some handicaps encountered by professionals in representing DEWATS in the design process. This question's answers helped us to understand some relationships between the use of certain project methods and how a specific BIM template could contribute to overcome these issues.

Sixth question's graph indications (Figure 8): More than half of the professionals considered the time spent working with DEWATS representation and the lack of adequate

information about these systems as handicaps. Almost half of the professionals found it difficult to find components on sale, such as CAD blocks or BIM components to represent DEWATS. Although there are many materials about these systems, for several professionals (almost 30%), the lack of materials on the subject was still a difficulty encountered by them.

### 3.6. Difficulties in Finding DEWATS BIM Components

The seventh question was aimed at understanding whether the difficulties in finding BIM templates for the representation of specific components, such as DEWATS, generate resistance to change from the traditional drafting CAD project method to BIM. The intention was to check if the lack of specific templates on sale was also a handicap in the transition from CADD to BIM methods and if the specific DEWATS template for BIM software could be another method that would help this transition.

Seventh question's graph indications (Figure 9): Most professionals (68%) "fully agreed" or "agreed" that the number of available BIM templates for sale can create resistance in the transition from CADD to BIM methods. On the other hand, 20% of the professionals did not know how to answer, whereas a small portion of these professionals (12%) "disagreed" or "totally disagreed" with this hypothesis.

## 4. Discussion

### 4.1. Conjectures on General CADD Software Users

In project design using general CADD software, it is common to use general drafting CAD blocks to represent project components as a way to increase productivity [26]. From the answers to the third question, we noticed that almost half of CADD software users (48%) still chose to use standard graphic representation blocks, although this is not the most suitable method for representing DEWATS, as each location has its particularities.

Establishing a parallel between these factors and the answers to the second question, it is possible to infer that many users tended to know only consolidated standard systems (such as septic tank, anaerobic filter, and soak pit) since there is great difficulty in finding drafting CAD blocks outside these normative standards, such as evapotranspiration tank, banana circle, constructed wetlands, vermifilter, and others. From the analysis of the answers to the sixth question, we believe that the long time spent in representing these systems in projects (pointed by 56% of the professionals) may be the reason why some of them end up making use of standards blocks. Moreover, it may lead them to consider these representations with tendencies from a "medium" to "low" level of development, as shown in the answers to the fourth question.

The method of drawing systems in general CAD software, answered by 40% of these users in the third question, led to the level of work for DEWATS characterization being higher in comparison with BIM software. Due to the need to redesign each system according to each project's characteristics, by using the BIM software, it is possible to configure a component with parameters that meet different design possibilities [25]. However, when it comes to secure understanding of the tool and work flexibility, general CADD software stands out [11], which may explain why that LOD had a more widespread distribution for these professionals, according to the answers to the fourth question, as it enables exploring several methods of project design.

This flexibility can also be related to the broad distribution of learning through design process of these users, as indicated in the answers to the fifth question. However, 40% of professionals drew DEWATS in projects according to their particularities, making learning somewhat unfeasible, since drawing these systems requires previous knowledge on this matter. This factor may have led most of them to answer that they did not learn about these systems throughout the design process (40%) or that they did not learn much (20%).

The transition from CADD to BIM methods represents a cultural change in project design, which might be one of the reasons for AECO professionals' resistance regarding BIM technology [11]. However, other factors that hinder this transition deserve due emphasis in the scientific field.

From the analysis of the answers to the sixth question, we noticed that more than half (56%) of the professionals considered the lack of information about DEWATS in software as a handicap in the design process. This factor can influence some professionals to look for these components on sale, where many of them also face difficulties. Thus, there may be a lack of motivation in the transition from CADD to BIM methods, not only for DEWATS documentation but for many other possibilities of project representation. This is one of the reasons why most professionals, according to the seventh question, also considered the lack of specific templates on sale as a handicap in the transition from CADD to BIM methods.

*4.2. Conjectures on BIM Software Users*

Many BIM software programs make DEWATS components and make them available for professionals to use in project modeling. Other BIM software programs give professionals the freedom to create these elements according to their particularities and preferences, and even allow them to sell these components, either within configured templates or separately. However, does the level of development of these pre-established components designed by the software or by other professionals meet the needs of the final users?

Analyzing the answers of the BIM software with generic template users to the fourth question, it is clear that there is a particular divergence of opinions. For almost half of them (48%), the chosen design method represented a perfect or satisfactory characterization of the system construction. For 52%, the chosen design method represented a regular or insufficient characterization for the system construction. This means that more than half of these professionals struggle to find DEWATS components, considered by them as having a high level of development. This factor may be why 1/5 of these professionals (20%) still develop standard geometries, draw the systems on general CADD software, or use general drafting CAD blocks to represent DEWATS in projects. The development of standard geometries or the use of standard CAD blocks are not suitable methods, as the first does not establish a sufficient level of development to understand the construction of the systems and the second, as stated above, has high probabilities of not meeting the needs of different regions.

BIM software allows the development of components to represent projects, the accommodation of their characteristics, and the establishment of parameters that respect construction standards. Hence, associating information in virtual models of DEWATS enables the professionals to choose individual systems, as in a catalog. Relating this factor to the answers to the second question, we believe that the use of BIM software to see more design alternatives has increased, in comparison to general CADD users, and a more specific BIM template leads to a higher tendency to learn more about these respective systems.

Given that the specific DEWATS template for BIM software is intended to be self-explanatory and educational, there is a possibility that the users of this software may learn more about DEWATS through the template itself. This hypothesis is ratified with the answers to the sixth question, showing that these professionals tended to learn a lot using this design method. However, it should also be noted that, because some professionals have sought a template that would meet their needs in the DEWATS representation, they may have a higher demand for this type of project and, consequently, are more acquainted with it.

Analyzing the answers to the third question, it can be seen that, considering all the BIM software users, whether with generic templates or with a specific DEWATS template, the tendency is to use pre-established BIM components, whether bought (68%), created by them (14%), or standard (8%), totaling 90% of these professionals. When a template or software has BIM components configured at a high LOD with several options, its users tend to use them in projects. This factor can be reinforced by relating these data to the answers to the fourth question, where it is clear that, in almost its entirety, BIM software with specific DEWATS template users answered that they consider the aid of component representations as having a "very high" LOD, and all of them use BIM components or bought templates. It

is worth mentioning that this hypothesis comes from the assumption that these users use specific template components themselves to represent DEWATS in projects.

*4.3. The Role of LOD on Learning (LODOL)*

It is possible to develop components within a BIM software and feed them with different types of information. This information may or may not be considered sufficient by professionals for the perfect characterization of a system on site. This factor is related to the virtual model's LOD. When a professional considers a BIM model's LOD very high, this indicates that, for them, that model presents all the details and information necessary for the budgeting, construction, operation, and maintenance of a particular system.

Analyzing the answers of BIM software with specific DEWATS template users, we noticed that almost all of them (84%) considered the available DEWATS components as having a very high LOD. However, 8% of the professionals considered it to be high, and the other 8% considered it to be medium. Hence, the consideration of specific levels of development is not unanimous by AECO professionals and may even be relative, although there is a tendency to consider the LOD of certain components as sufficient to build a system. Hence, what would this information be, and how could it be measured?

Researching the BIMForum's level of development specification, one realizes that only one item (sanitary sewerage equipment) can be assigned to the septic tank, where the LOD is rated from 100 to 400. In Brazil, there are two project presentation books in BIM, from Parana and Santa Catarina. In the first, only the elements "septic tank" and "soak pit" are mentioned, showing the details and information that these components must have to reach the Level of Detail and Information 3, as established by them. In the second, DEWATS are not mentioned. In other words, in the BIMForum's specification and in the two available Brazilian notebooks dealing with LOD specifications, adequate information is not found for most of these systems.

Another point to note is that, even though there are many teaching materials about DEWATS in the literature (several of them are cited in this paper), one realizes that almost 30% of professionals still find it difficult to access these materials. Thus, we argue whether BIM could be an allied tool in disseminating knowledge about these systems. In the BIM software, there is the possibility of associating information in virtual models and creating catalogs of DEWATS options. The professionals can come across systems not known by them, analyze the relevant information, and consequently get to know their applications and features. It may be that the professional's level of learning about these systems is related to its level of detail and information.

When the model is presented with an LOD of 400, it means that it contains sufficient detail and accuracy for the represented component fabrication. An LOD of 400 is the maximum level that a BIM model can reach in the design process. Although an LOD of 500 is above an LOD of 400, it involves as-built generation and takes place at the end of the construction management phases, that is, after the design stage. With that said, analyzing the answers to the fifth question and comparing them with the answers to the fourth question, it is clear that a higher LOD estimated by the professionals reflects a greater openness to learning about DEWATS through the design tool. That is, there may be a strong relationship between the model's level of development and how much professionals can learn by working with this system in the design process, as we can see in Figure 10.

Therefore, we produced Table 3 to present a way to measure the amount of information and details that DEWATS must have in order to advance from an LOD of 100 to an LOD of 500. In this table, a new LOD is inserted: a level of development on learning (LODOL) or LOD of 450. At this level, the model presents enough detail and information to favor learning about these systems in the design process itself. That is, it proposes the virtualization of information in the learning process. This table suggests the information and details that the DEWATS models should present for specific application contexts, relating them to a defined LOD. This information may disseminate knowledge about DEWATS,

assisting professionals with the various possibilities of applications and contributing to the sustainable implementation of these systems.

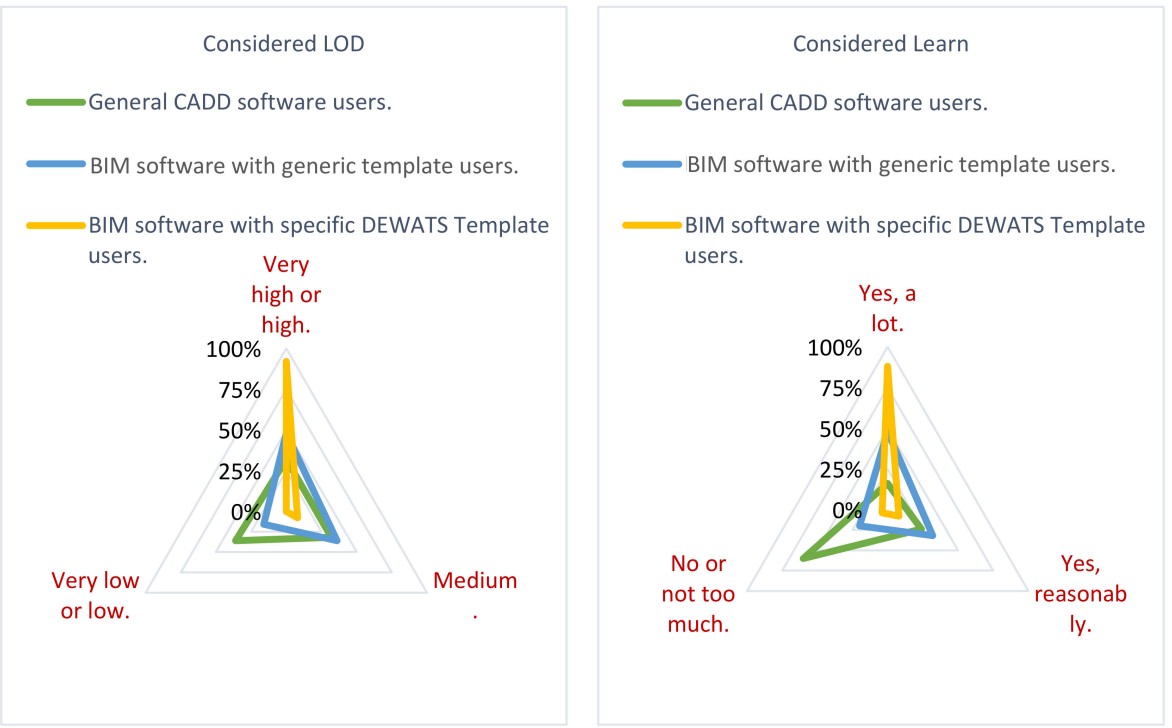

**Figure 10.** Comparison between answers to the fourth (**left**) and fifth (**right**) questions. Source: authors (2020).

**Table 2.** Level of development (LOD) on learning specifications for DEWATS.

| Level of Development | Information | Details | Application Context |
|---|---|---|---|
| LOD 100 | Data of the place where it will be implanted:<br>● Region of implantation;<br>● Building occupation;<br>● Number of contributors;<br>● Average local temperature;<br>● Type of soil;<br>● Depth of the water table. | Two-dimensional generic symbols or illustrations.<br> | Initial design studies. |
| LOD 200 | Materials and typologies definitions:<br>● Materials (reinforced concrete, masonry, and others);<br>● Typology (molded on site or prefabricated).<br>● Standard dimensions;<br>● Location;<br>● Guidance;<br>● System performance analysis. | Generic geometry with flexible dimensions.<br> | Product definition. |
| LOD 300 | Initial sizing:<br>● Sizing data (eviction contribution, flow, holding period, and others)<br>● Minimum dimensions required (calculated minimum volume, calculated minimum area, and others). | Geometry with defined general and specific dimensions.<br> | Interface identification and solution. |

**Table 3.** *Cont.*

| Level of Development | Information | Details | Application Context |
|---|---|---|---|
| LOD 350 | Final sizing:<br><br>• Configuration of the system's elements;<br>• Material quantitation;<br>• System's real dimensions (internal and external concrete and pipe diameters, ceramic blocks dimensions, real length, real width, and others);<br>• Interface with other systems;<br>• Descriptive memorials of calculation. | System element representation.<br><br> | Interface identification and solution. |
| LOD 400 | Final details and specifications:<br><br>• System element specifications (brand, model, manufacturer, price, description, and others);<br>• Constructive methods (step-by-step for construction, materials used, planning, and others);<br>• Preventive and corrective operation and maintenance information. | Detailing required for system manufacturing, assembly, and installation.<br><br> | Specialization detail project. |
| LODOL or LOD 450 | Learning about systems:<br><br>• Relevant standards (items and tables);<br>• Sizing methods (formulas, units, data identity and tests);<br>• Functionality (how the system works);<br>• Applications (type of system, type of treated sewage, treatment stage, and others);<br>• Recommendations (tips and guidelines);<br>• Considerations (rate of removal of organic matter, frequency of maintenance, curiosities, and others);<br>• References (articles, books and documents). | | Learning. |
| LOD 500 | As-built:<br><br>• Date of component purchase and installation;<br>• Installation records and how it was built. | As-built details. | Delivery of the work. |

Source: authors (2020).

## 5. Conclusions

This observational study's conclusions provide crucial data on the experience with the use of BIM in teaching contexts, providing the necessary provisional guidelines for this type of use in practice. These findings support recent studies presented in the theoretical framework, indicating a potential role for LOD in the constructive communication of a system and in providing information that allows learning throughout the design process. These are interesting results considering the importance of DEWATS in the current environmental context to promote public health, knowing that many professionals do not know the various possibilities of these systems, and noticing that works aimed at the development of design technologies for these types of systems are still scarce. It is then realized the necessity to explore tools of design technologies that contribute to the dissemination of knowledge about DEWATS. However, the LODOL proposal presented in this paper should be interpreted with some caution and should not be applied interchangeably, because it does not replace any teaching/learning material on the subject, but rather contributes as one more method of knowledge dissemination.

**Author Contributions:** Conceptualization, M.A.D. and A.A.; methodology, M.A.D. and A.A.; software, M.A.D.; validation, M.A.D.; formal analysis, M.A.D.; investigation, M.A.D.; resources, M.A.D.; data curation, M.A.D.; writing—original draft preparation, M.A.D. and A.A.; writing—review and editing, M.A.D. and A.A.; visualization, M.A.D.; supervision, M.A.D. and A.A.; project administration, M.A.D.; funding acquisition, M.A.D. and A.A. All authors have read and agreed to the published version of the manuscript.

**Funding:** This study was financed in part by the Coordenação de Aperfeiçoamento de Pessoal de Nível Superior—Brasil (CAPES)—Finance Code 001.

**Institutional Review Board Statement:** Not applicable.

**Informed Consent Statement:** Not applicable.

**Data Availability Statement:** The results obtained in the questionnaire can be viewed through the following link: https://docs.google.com/forms/d/1pM30ZzYOTozXQMbfhzfh-oSp7cqohcWGBx4_FAKg4BM/viewanalytics (accessed on 5 May 2021).

**Acknowledgments:** The authors would like to thank the Coordenação de Aperfeiçoamento de Pessoal de Nível Superior—Brasil (CAPES) for financially supported this study. We gratefully acknowledge the Water editors, publication team at MDPI, and anonymous reviewers for their professional comments and efforts, which greatly improved this manuscript.

**Conflicts of Interest:** The authors declare no conflict of interest.

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
