# Peer review of "The Potential of BIM in Disseminating Knowledge about Decentralized Wastewater Treatment Systems: Learning through the Design Process"

_water, doi:10.3390/w13111504_

Round 1
Reviewer 1 Report
This manuscript makes an important contribution to the sanitation literature. It is good to see how BIM concepts can be used as learning tools.
One comment is that while the manuscript is well written, the authors could restructure the presentation of their results. I think it would be helpful if the results could be presented under the various themes/questions. It would also be interesting to see the difference in responses from the three groups of respondents (line 20). This would certainly ensure clarity.
Author Response
Point 1: One comment is that while the manuscript is well written, the authors could restructure the presentation of their results. I think it would be helpful if the results could be presented under the various themes/questions. It would also be interesting to see the difference in responses from the three groups of respondents (line 20). This would certainly ensure clarity.
Response 1: Dear Reviewer,
Thank you very much for reviewing my survey, your tips were essential for me to restructure it. Regarding your comments, I restructured the results under the various themes, showing the reason why I asked the certain questions. In line 20, I indicated the groups and the differences. Any other suggestions, I am at your disposal. I am so gratefull.
Reviewer 2 Report
The article is interesting and deals with very important issues. However, I have a problem with writing a review of this article because the essence of the article actually refers to the BIM design system, which is used very widely for the design of various structures. In this case it refers to waste water treatment systems, but only in a reference sense only. Please note that the article was submitted to the Water Journal, and the relationship between the article and water, here with wastewater, is very loose. In wastewater treatment systems, the basis is the process technology and further the efficiency of wastewater treatment, the balance of pollutant loads, the intensity of aeration, wastewater mixing, retention times, and this is the basis for selecting devices and constructing a technological line. The problem posed in this way would be consistent with the topic of the Water Journal.
I have a few more detailed comments.
- Figure 1 - It's nice, but not sure what each device means.
- Fig. 2 - what is the technological relationship with the presentation of the design specifying the dimensions of the concrete tank and pipes?
- Figures 5 and 6 need clarification. I don't understand them.
- Table 2 - in the figure in LOD 450, the descriptions are not in English.
- Descriptions should be standardized in the references. The year of publication is not written uniformly.
The questionnaire research is an undoubted achievement, but it is rather in the field of sociological research. If the article is to be published in Water, it should be supplemented with information more explicitly related to wastewater treatment than to general BIM design of various structures.
Author Response
Point 1: The article is interesting and deals with very important issues. However, I have a problem with writing a review of this article because the essence of the article actually refers to the BIM design system, which is used very widely for the design of various structures. In this case it refers to waste water treatment systems, but only in a reference sense only. Please note that the article was submitted to the Water Journal, and the relationship between the article and water, here with wastewater, is very loose. In wastewater treatment systems, the basis is the process technology and further the efficiency of wastewater treatment, the balance of pollutant loads, the intensity of aeration, wastewater mixing, retention times, and this is the basis for selecting devices and constructing a technological line. The problem posed in this way would be consistent with the topic of the Water Journal.
The questionnaire research is an undoubted achievement, but it is rather in the field of sociological research. If the article is to be published in Water, it should be supplemented with information more explicitly related to wastewater treatment than to general BIM design of various structures.
Response 1: Dear Reviewer,
Thank you very much for reviewing my survey, your tips were essential for me to restructure it. Regarding your comment, I made some changes to adapt the article with greater focus on DEWATS. The purpose of this article was to verify that BIM tools can contribute to the dissemination of knowledge about these systems, helping professionals to make the best choices. I apologize if it was not very clear.
In the summary topic, I made some changes to indicate this goal in line 17, 18,19, 20, and 21.
In the methodology topic, I added some information in Figure 2, added more information in lines 193-198, and a new figure (Figure 3) showing how the developed Template can assist professionals in choosing the best DEWATS for certain situations.
In the discussion topic, I added new information (line 485-487) showing possible contributions to the implementation of DEWATS.
Point 2: Figure 1 - It's nice, but not sure what each device means.
Response: I enlarged the image, but I believe that it would not be fitting to put the name of each system in the image, it would be difficult to see. The intention of the image was only to show the graphic representation of the systems that were mentioned in lines 173-177.
Point 3: Fig. 2 - what is the technological relationship with the presentation of the design specifying the dimensions of the concrete tank and pipes?
Response 3: This relationship with the dimensions of the materials used allows professionals to assess quantitative and budgetary issues for the implementation of DEWATS.
Point 4: Figures 5 and 6 need clarification. I don't understand them.
Response 4: I changed some colors for better visualization, but these are radar graphics. Each tip on the graph indicates a percentage of the response for a particular group.
Point 5: Table 2 - in the figure in LOD 450, the descriptions are not in English.
Response 5: I changed the figure for one with English descriptions.
Point 6: Descriptions should be standardized in the references. The year of publication is not written uniformly.
Response 6: I changed the description of some references trying to make them uniform (lines 563, 564, 569, and 594)
Any other suggestions, I am at your disposal. I am very grateful for all the comments.
Reviewer 3 Report
Title: The potential of BIM in disseminating knowledge about Decentralized Wastewater Treatment Systems: learning through the design process
- check for spaces; there seem several space issues in the manuscript.
- this is a research paper but looks like a review paper and the authors did not follow the manuscript preparation!
- Line numbers are wrong, so it is difficult to read and follow and comment on the specific sentences.
- The introduction is very general and long! It should summarize the key research gap in this area and how this review can help researchers to overcome the issues.
- The conclusion should be very brief, not more than 8-12 lines.
- It's better to change fig 6 graph to make it more clear!
Author Response
Point 1: check for spaces; there seem several space issues in the manuscript.
Response 1: Dear Reviewer,
Thank you very much for reviewing my research, your tips were essential so that I could restructure it. I checked the space issue and adjusted them all!
Point 2: this is a research paper but looks like a review paper and the authors did not follow the manuscript preparation!
Response 2: I made some changes to the methodology in order to clarify the method used and how the artifact was developed (lines 190-201). We have developed an artifact based on the Design Science Research method and the adoption of qualitative analysis of a questionnaire answered by three groups of professionals allowed us to identify differences among then. I don't understant how it would be a review paper.
Point 3: Line numbers are wrong, so it is difficult to read and follow and comment on the specific sentences.
Response 3: I checked the line number issue and adjusted them all!
Point 4: The introduction is very general and long! It should summarize the key research gap in this area and how this review can help researchers to overcome the issues.
Response 4: I summarized a good part of the introduction in topic 1.3 - "BIM as a learning tool". The gap is found on lines 64-74 where we show that not many technologies have been researched for DEWATS projects and that many professionals are not aware of the various DEWATS possibilities that can be used. And to overcome the issues, we can improve project technologies that favor the learning of professionals about these systems, helping in the correct choice (lines 130-139).
Point 5: The conclusion should be very brief, not more than 8-12 lines.
Response 5: I summarized a good part of the conclusion
Point 6: It's better to change fig 6 graph to make it more clear!
Response 6: I changed the colors of the texts and increase the font. It is a radar chart where the points represent the percentage of response from each group.
Any other suggestions, I'm at your disposal . I am very grateful for the comments. Thank you very much!
Round 2
Reviewer 2 Report
I think that the text can be accepted for publication in its current form. It is definitely closer to the magazine's profile.
I have one suggestion.
There are various objects in Fig. 1, but many people will not know what they are for. Some of them, it seems to me, belong to common groups. You can learn a little about some of them later in the article, because only from Table 2. It would be worth, perhaps using arrows, to show at least these groups and name them.
Reviewer 3 Report
Thank you